LPHE-MS-June-2024

Prepared for submission to JHEP

# Higher spin swampland conjecture for massive AdS$_3$ gravity

**R. Sammani,** [1,2] **E.H Saidi**[1,2]

[1]*LPHE-MS, Science Faculty, Mohammed V University in Rabat, Morocco*

[2]*Centre of Physics and Mathematics, CPM- Morocco*

*E-mail:* sammani.rajaa@gmail.com, saidieh@gmail.com

ABSTRACT: In this paper, we show that a possible version of the swampland weak gravity conjecture for higher spin (HS) massive topological AdS$_3$ gravity can be expressed in terms of mass $\mathrm{M_{hs}}$, charge $\mathrm{Q_{hs}}$ and coupling constant $g_{\mathrm{hs}}$ of 3D gravity coupled to higher spin fields as $\mathrm{M_{hs}} \leq \sqrt{2}\mathrm{Q_{hs}}g_{\mathrm{hs}}M_{\mathrm{Pl}}$. The higher spin charge is given by the $SO(1,2)$ quadratic Casimir $\mathrm{Q_{hs}^2} = s(s-1)$ and the HS coupling constant by $g_{\mathrm{hs}}^2 = 2/\left(M_{\mathrm{Pl}}^2 l_{\mathrm{AdS_3}}^2\right)$ while the mass expressed like $(l_{\mathrm{AdS_3}}\mathrm{M_{hs}})^2$ is defined as $(1 + \mu l_{\mathrm{AdS_3}})^2 s(s-1) + [1 - (\mu l_{\mathrm{AdS_3}})^2 (s-1)]$.

KEYWORDS: Higher spin BTZ black hole, Swampland conjecture for higher spin AdS$_3$, gravity, Higher spin topological massive gravity, SL(2,$\mathbb{R}$) representations.

# 1 Introduction

With the purpose of investigating and deriving possible swampland constraints [1]-[3] on topological gravity [4, 5] coupled to higher spin massive fields, we consider the three dimensional AdS action with a negative cosmological constant in addition to a gravitational Chern-Simons (CS) term to build up the higher spin topological massive gravity (HSTMG) theory [6]-[9]. Particularly, we are interested in the higher spin Bañados-Teitelboim-Zanelli (BTZ) black hole solution [10, 11] and its discharge.

This study is driven by several motives, mainly the non-supersymmetric AdS [12, 13] conjecture and the weak gravity constraint [14–16]. In fact, it was stipulated that non supersymmetric AdS spaces—as well as locally lookalike AdS geometries— are at best metastable; they manifest a non perturbative instability and will ultimately decay [2, 3]. And since the BTZ black hole is locally isometric to AdS$_3$, it should also a priori exhibit a similar instability [4].

Actually, black holes in AdS spaces are of two types [17]: we either have large black holes in equilibrium with their thermal bath, or small unstable black holes in need of discharging by radiating away their charge. This aligns with the weak gravity conjecture,

which requires the emission of a super-extremal particle with a constraint on its mass to charge ratio. However, it was argued in [17] that this mild version of the WGC, demanding a single super-extremal state, is not sufficient and a lattice refinement of the constraint is more suitable. To insure the decay of BTZ black holes in AdS$_3$, one must guarantee that the emitted particles reach the AdS$_3$ boundary and don't bounce back to form a self-interacting particle condensate that could eventually become sub-extremal [17]. Indeed, the boundary conditions on the AdS$_3$ cylinder can box the discharged particles enabling them to interact in a sub-extremal cloud of emitted particles. Therefore, one must require instead a stronger version of the weak gravity conjecture, a lattice WGC, where in each charge sector, there should be a super-extremal state [17].

For unstable BTZ black holes in HSTMG, and in order to comply the super-extremality constraint of the WGC, there must be a set of emitted super-extremal particles that ought to be charged and massive higher spin particles. Now, is it possible to formulate such WGC constraint for higher spin topological massive gravity to regulate the discharge of unstable higher spin BTZ black holes?

To the best of our knowledge, this inquiry was never investigated in Literature. The WGC constraint was only established for a disjointed setting, where the gravitational and gauge sectors are separated by considering 3D gravity in addition to a U(1) gauge field [17, 18] but never for massive AdS$_3$ gravity in CS formulation coupled to higher spin fields.

In this paper, we intend to fill in this gap by first reviewing known results on the D-dimensional black holes and their WGC constraints to formulate our hypothesis about the expected super-extremality bound for the HS-BTZ black hole (section 2). Then, we construct the mass and charge operators to build the higher spin states (section 3). Once we have all the tools needed, we derive the swampland constraint for higher spin BTZ black holes and compute the tower of super-extremal higher spin states (section 4). And before concluding, we discuss the relevance of our findings with regard to recent progress in the swampland program Literature (section 5).

## 2   Weak gravity conjecture in D $\geq$ 4

This section aims to motivate a Swampland conjecture for higher spin massive AdS$_3$ gravity to regulate the discharge of higher spin BTZ black holes using commonly accepted arguments. To pave the way for this 3D Swampland constraint, we intend to align it with the weak gravity conjecture (WGC) governing the decays of black holes

in space dimensions $D > 3$ [14]. This bridging between HS- AdS$_3$ models and effective gauge theories coupled to D-gravity (EFF$_D$) is sustained by several facts and features; in particular:

(**A**) the existence of a Chern-Simons (CS) gauge formulation of higher spin AdS$_3$ gravity [19, 20]. In this formalism, one uses standard gauge fields valued in the Lie algebra of the CS gauge symmetry $\mathcal{G}_{hs} \times \tilde{\mathcal{G}}_{hs}$; this, as we will see, permits to replicate the construction of certain constraints regulating the decay of D-black holes for unstable HS-BTZ black holes. For the remainder of this investigation, we focus on higher spin BTZ black hole solutions in the CS formulation with rank 2 gauge symmetries namely: (**1**) the higher spin SL(3,$\mathbb{R}$) model [21] having two spins $s = 2, 3$ as a representative of the HS theory with SL(N,$\mathbb{R}$) family [22]. (**2**) The higher spin model with SO(2,3) group having also two spins $s = 2, 4$ as a representative of the HS ortho-symplectic families with gauge symmetries given by the real split forms of B$_N$, C$_N$ and D$_N$ Lie groups [23]. And (**3**) the exceptional G$_2$ higher spin model [24] with spin spectrum given by $s = 2, 6$. As well, this G$_2$ can be viewed as a representative of the exceptional family of finite dimensional Lie algebras. Useful characteristic properties of these HS topological gauge models are as follows

| symmetry $\mathcal{G}_{hs}$ | spin set $J_{\mathcal{G}}$ | generators | dim $\mathcal{G}_{hs}$ |
|---|---|---|---|
| SL(3,$\mathbb{R}$) | 2, 3 | $W_{m_1}^{(1)} \oplus W_{m_2}^{(2)}$ | $3 + 5$ |
| SO(2,3) | 2, 4 | $W_{m_1}^{(1)} \oplus W_{m_3}^{(3)}$ | $3 + 7$ |
| G$_2$ | 2, 6 | $W_{m_1}^{(1)} \oplus W_{m_5}^{(5)}$ | $3 + 11$ |

$$(2.1)$$

with label $m_j$ taking integral values as $-j \leq m_j \leq j$. The $W_{m_j}^{(j)}$ are the generators of the spin $s_j = j + 1$; they form an isospin j representation of the principal sl(2;$\mathbb{R}$) partitioning the $\mathcal{G}_{hs}$ generators as exhibited by the two last columns of the above table.

The second feature supporting the EFT$_D$-HS AdS$_3$ cross over is (**B**) the AdS$_3$/CFT$_2$ correspondence [28] allowing to relate topological aspects of HS- AdS$_3$ gravity such as Wilson lines with conformal highest weight representations and conformal observables; which will be fundamental for the computation of the HS Swampland constraint. And lastly, the possibility to (**C**) realise both the masses and the charges required by the WGC in terms of the quantum numbers of the CS gauge symmetry, as well as the coupling constants and the Planck mass M$_{Pl}$. The EFT$_D$-HS AdS$_3$ crossing is therefore based on matching the D- dimensional WGC ingredients with those of 3D higher spin gravity as

follows:

| $D > 3$ | $D = 3$ |
|---|---|
| D-gravity + U(1) charged matter | 3D gravity coupled to higher spin fields |
| Effective field models | Higher spin Chern-Simons formulation |
| Electrically charged Black holes | Higher spin BTZ black holes |
| Electric charge $q_e$ | Spin charge $Q_{hs}$ |
| Mass $\boldsymbol{m}$ | Mass of HS particles $M_{hs}$ |

$$(2.2)$$

We show throughout this paper that the emitted particle states $|s; \{\lambda\}\rangle$ of the 3D BTZ black hole carry, in addition to the higher spins s, masses $M_{hs}$ and charges $Q_{hs}$ which are functions of s. These states will be denoted below like

$$|s; M_{hs}, Q_{hs} > \tag{2.3}$$

where the masses $M_{hs}$, and the charges $Q_{hs}$ are eigenvalues of some function of commuting observables $\mathcal{O}_i$ of the gauge theory with symmetry group $\mathcal{G}_{hs} \times \tilde{\mathcal{G}}_{hs}$. Candidates for these $\mathcal{O}_i$s are given by the Cartan charge operators of $\mathcal{G}_{hs} \times \tilde{\mathcal{G}}_{hs}$ and their Casimirs.

In this regard, we restrict to the principal SL(2,$\mathbb{R}$) symmetry observables within the gauge symmetry $\mathcal{G}_{hs}$; they are given by the Cartan charge $L_0$ and the quadratic Casimir $\mathcal{C}_2$ with the following commutation relations

$$[L_n, L_m] = (m - n) L_{n+m} \tag{2.4}$$

$$\mathcal{C}_2 = L_0^2 - L_0 - L_+ L_- \tag{2.5}$$

Particularly, we are interested in the mass $\hat{M}_{hs}$ and the charge $\hat{Q}_{hs}$ operators with spectrums as follows

$$\begin{aligned} M_{hs} &:= \text{spect}\left(\hat{M}_{hs}\right) \\ Q_{hs} &:= \text{spect}\left(\hat{Q}_{hs}\right) \end{aligned} \tag{2.6}$$

They can be expanded in terms of the commuting Cartan charge $L_0$ and the Casimir $\mathcal{C}_2$ of the principal SL(2,$\mathbb{R}$) symmetry like

$$\hat{M}_{hs}^2 = \mathfrak{m}_0 L_0 + \mathfrak{m}_2 \mathcal{C}_2 \tag{2.7}$$

$$\hat{Q}_{hs}^2 = \mathfrak{q}_0 L_0 + \mathfrak{q}_2 \mathcal{C}_2 \tag{2.8}$$

with some positive $\mathfrak{m}_i$ and $\mathfrak{q}_i$ to be determined later on. The mass $\hat{M}_{hs}$ and charge $\hat{Q}_{hs}$ observable operators act on the higher spin-s particle $|s; M_{hs}, Q_{hs} >$ states as

$$\begin{aligned} \hat{M}_{hs}|s; M_{hs}, Q_{hs} > &= M_{hs}|s; M_{hs}, Q_{hs} > \\ \hat{Q}_{hs}|s; M_{hs}, Q_{hs} > &= Q_{hs}|s; M_{hs}, Q_{hs} > \end{aligned} \tag{2.9}$$

Before proceeding any further, we pause to carefully examine and comment on the structure of the operators (2.7-2.8) by leveraging well-known principles from the $AdS_3/CFT_2$ correspondence [25, 26] and $SL_2$ isospin representation framework:

(**1**) for the $\hat{M}_{hs}^2$ expansion (2.7), the block term generated by $L_0$ can be attributed to the $CFT_2$ relationship $m \sim h + \bar{h}$ with the eigenvalue equations $L_0 |h\rangle = h |h\rangle$ and $\bar{L}_0 |\bar{h}\rangle = \bar{h} |\bar{h}\rangle$. Regarding the block term generated by $C_2$, it can be motivated by the Sugawara construction of the conformal energy momentum tensor from the affine $SL_2$ Kac-Moody current [27].

(**2**) As for the expansion of the operator $\hat{Q}_{hs}$ in (2.8), it can be restricted to the Casimir block $\hat{Q}_{hs}^2 = q_2 C_2$ with some $q_2 > 0$. This is because the Casimir operator $C_2$ captures information on the $SL_2$ isospin $\Delta$ while the charge operator $L_0$ (thought of as $J_z$) captures data on the isospin projection $\Delta_z$.

Taking all of the aforementioned into account, one might speculate that the mass $\hat{M}_{hs}$ and the charge $\hat{Q}_{hs}$ operators are indeed linked to each other like

$$\hat{M}_{hs}^2 = \mathfrak{m}_0 L_0 + \frac{\mathfrak{m}_2}{\mathfrak{q}_2} \hat{Q}_{hs}^2 \tag{2.10}$$

Such property justifies the interest in the search for a Swampland conjecture for higher spin gravitational models. Moreover, by acting on the quantum states $|\Delta, N\rangle$ of (unitary) representations $\mathcal{R}_\Delta^\pm$ of the $SL(2,\mathbb{R})$ symmetry group with both sides of the above equation, we get the following mass relation

$$\begin{aligned} \mathcal{R}_\Delta^+ &: \quad M_{\Delta,N_+}^2 &=& \quad +\mathfrak{m}_0 (\Delta + N) + \tfrac{\mathfrak{m}_2}{\mathfrak{q}_2} \Delta (\Delta - 1) \\ \mathcal{R}_\Delta^- &: \quad M_{\Delta,N_-}^2 &=& \quad -\mathfrak{m}_0 (\Delta + N) + \tfrac{\mathfrak{m}_2}{\mathfrak{q}_2} \Delta (\Delta - 1) \end{aligned} \tag{2.11}$$

The structure of the representations $\mathcal{R}_\Delta^\pm$ and the properties of the states $|\Delta_\pm, N_\pm\rangle$ will be thoroughly investigated in *subsection 3.1.* Meanwhile notice that the set of HS quantum states $|s; M_{hs}, Q_{hs} >$ has a group theoretic basis; they can be perceived as the $|\Delta, N\rangle$ of the $SL_2$ representation group theory which will be proven to be accurate.

Returning to the HS Swampland conjecture issue, we seek to show that the decay of small HS- BTZ black holes in $AdS_3$ gravity is accompanied by the emission of super-extremal higher spin-s states $|s; M_{hs}, Q_{hs} >$ with spin dependent masses $M_{hs}$ and charges $Q_{hs}$ constrained as follows

$$M_{hs} \leq \sqrt{2} Q_{hs} g_{hs} M_{Pl} \tag{2.12}$$

with $g_{\text{hs}}$ standing for the higher spin coupling constant to be determined later. Below, we refer to (2.12) as the Swampland higher spin conjecture (**HSC**) for massive AdS$_3$ gravity. At first impression, one might wonder about the interpretation of such constraint and whether this inequality is a true swampland conjecture. However by way of construction, the swampland HSC will prove to be a version of the WGC that regulates the discharge of higher spin BTZ back hole solutions of HSTMG carrying charges beyond the usual U(1) of [17]. The HSC accounts for 3D black holes solutions with different backgrounds than the ones already considered in Literature [17, 18], and can be therefore perceived as a complement to the work conducted in AdS$_3$ framework regarding the derivation of the WGC.

Moreover, the condition (2.12) has interesting properties shared by unstable D-black holes. Particularly, the constraint (2.12) has a quite similar structure to the well known 4D weak gravity conjecture formulated by the following inequality [14]

$$\boldsymbol{m} \leq \sqrt{2} q g_{U(1)} M_{\text{Pl}} \tag{2.13}$$

This well established constraint relation (2.13) will be used as a guiding principle for the derivation of the **HSC** (2.12). To avoid confusion between the 3D and 4D parameters, we use the following convention notations

| black hole | mass | charge | coupling |
|---|---|---|---|
| 4D charged BH | $\boldsymbol{m}$ | $q$ | $g_{U(1)}$ |
| 3D BTZ | $M_{\text{hs}}$ | $Q_{\text{hs}}$ | $g_{\text{hs}}$ |

$$\tag{2.14}$$

In dimensions $D \geq 4$, the weak gravity conjecture (WGC) requires the existence of at least one super extremal state $|\boldsymbol{m}, q\rangle$ in the particle spectrum of the effective U(1) gauge theory coupled to D- gravity with mass $\boldsymbol{m}$ and charge $Q_{U(1)} = q g_{U(1)}$ satisfying the condition [2, 14]

$$q^2 g_{U(1)}^2 \geq \frac{D-3}{D-2} \boldsymbol{m}^2 M_{\text{Pl}}^{2-D} \tag{2.15}$$

where the gauge coupling constant $g_{U(1)}$ scales like MASS$^{2-D/2}$ and $M_{\text{Pl}}$ is the D- Planck mass. This constraint relation puts a condition on the allowed space time dimensions as it requires $D \geq 3$; although the $D = 3$ is a critical value. By putting D=3, the relation (2.15) leads to a trivial condition $q^2 g_{U(1)}^2 \geq 0$ with no reference whatsoever to the value of the mass $\boldsymbol{m}^2$. Even with a reverse reasoning, if we consider instead $\boldsymbol{m}^2 \leq q^2 g_{U(1)}^2 M_{\text{Pl}}^{D-2} (D-2)/(D-3)$, all we learn is that $\boldsymbol{m}^2 \leq \infty$ lacking any information on the value of $q^2$.

However, to retrieve additional insights, we concentrate on the interesting four dimensional theory ($D = 4$). The condition (2.15) reads like

$$\boldsymbol{m}^2 \leq 2q^2 g_{U(1)}^2 M_{\mathrm{Pl}}^2 \tag{2.16}$$

with

$$q = \int_{\mathbb{S}^2} \mathbf{E}_{u(1)}.d\boldsymbol{\sigma} \tag{2.17}$$

and where $\mathbf{E}_{u(1)} = -\boldsymbol{\nabla} V - \partial_t \mathbf{A}$ is the usual electric field. For this U(1) abelian gauge theory, gauge group elements $\mathcal{U}$ are given by $e^{i\boldsymbol{Q}_{u(1)}}$ with generator

$$\boldsymbol{Q}_{u(1)} = g_{u(1)}\mathfrak{Q} \tag{2.18}$$

acting on charged quantum states like

$$\boldsymbol{Q}_{u(1)} \ket{\boldsymbol{m}, q} = q g_{u(1)} \ket{\boldsymbol{m}, q} \qquad , \qquad \hat{\boldsymbol{M}}^2 \ket{\boldsymbol{m}, q} = \boldsymbol{m}^2 \ket{\boldsymbol{m}, q} \tag{2.19}$$

where $\hat{\boldsymbol{M}}$ is the mass operator. In the upcoming section, we construct the higher spin homologue of (2.19) for HS-AdS$_3$ gravity.

# 3 Higher spin particle states

An essential key component to the derivation of the relation (2.12), is the set of emitted super extremal particle states $\ket{s; \mathrm{M_{hs}, Q_{hs}}} >$ . It is therefore crucial, before all else, to define these states. We identify these particles as eigenstates of some higher spin charge operator defined like $\boldsymbol{Q}_{\mathrm{hs}} = g_{\mathrm{hs}} Q$ analogously to the 4D charge operator $\boldsymbol{Q}_{u(1)} = g_{u(1)}\mathfrak{Q}$ given by eq(2.18). It acts as follows

$$\boldsymbol{Q}_{\mathrm{hs}} \ket{s; \mathrm{M_{hs}, Q_{hs}}} = g_{\mathrm{hs}} \mathrm{Q_{hs}} \ket{s; \mathrm{M_{hs}, Q_{hs}}} \tag{3.1}$$

where $g_{\mathrm{hs}}$ is the higher spin coupling constant of the higher spin gauge theory, it will be computed later on [ see eq(4.12)].

To manoeuvre the set of these states, we use the principal SL(2,$\mathbb{R}$) representations since all the rank 2 gauge symmetries $\mathcal{G}_{\mathrm{hs}} \times \tilde{\mathcal{G}}_{\mathrm{hs}}$ we are considering can be obtained via the principal embedding of SL(2,$\mathbb{R}$). Therefore, we deem it necessary to briefly recall results on the principal SL(2,$\mathbb{R}$) subgroup of the gauge symmetry $\mathcal{G}_{\mathrm{hs}}$ and its unitary representations.

### 3.1 Unitary representations of SL(2,ℝ)

SL(2,ℝ) is a non compact group homomorphic to the Lorentz SO(1,2) and generated by $L_0$, $L_\pm$ with commutation relations $[L_n, L_m] = (m-n)L_{n+m}$ labelled by $n, m = 0, \pm$. It has several families of irreducible representations that can classified into two sets [30, 31], non unitary and unitary. The latter will be the focus of the upcoming discussion.

Unitary irreducible representations (UIR) are infinite dimensional, they are obtained by requiring the hermiticity condition $L_n^\dagger = L_{-n}$ and the positivity of the quantum states norms; i.e: $\||\psi>\| > 0$. An interesting type of these UIRs is given by the discrete series denoted like $\mathcal{R}_\Delta^\pm$ [29–31]:

(**1**) Discrete series $\mathcal{R}_\Delta^+$ are generated by the quantum states $|\Delta, N\rangle$ as follows

$$\begin{aligned}
L_+ |\Delta, N\rangle &= \sqrt{(N+1)(N+2\Delta)} \, |\Delta, N+1\rangle \\
L_- |\Delta, N\rangle &= \sqrt{N(N+2\Delta-1)} \, |\Delta, N-1\rangle \\
L_0 |\Delta, N\rangle &= (N+\Delta) \, |\Delta, N\rangle \\
\mathcal{C}_2 |\Delta, N\rangle &= \Delta(\Delta-1) \, |\Delta, N\rangle
\end{aligned} \tag{3.2}$$

where $\mathcal{C}_2$ is the SL(2,ℝ) quadratic Casimir $L_0^2 - L_0 - L_+ L_-$. From these relations, one can compute useful quantities to draw several properties; in particular:

(**i**) the norm $< \Delta, N|L_+L_-|\Delta, N >$ which is equal to $N(N+2\Delta-1)$. And its homologue $< \Delta, N|L_-L_+|\Delta, N >$ given by $(N+1)(N+2\Delta)$.

(**ii**) The representation $\mathcal{R}_\Delta^+$ is bounded from below indicating that $L_- |\Delta, N\rangle = 0$ and requiring therefore $N(N+2\Delta-1) = 0$.

This latter constraint can be solved for $N = 0$, and the state $|\Delta, 0\rangle$ with positive definite $\Delta$ is thus a lowest weight state obeying the following lowest weight relations

$$\begin{aligned}
L_- |\Delta, 0\rangle &= 0 \\
L_0 |\Delta, 0\rangle &= \Delta |\Delta, 0\rangle \\
\mathcal{C}_2 |\Delta, 0\rangle &= \Delta(\Delta-1) |\Delta, 0\rangle
\end{aligned} \tag{3.3}$$

With $L_+$ acting on $|\Delta, 0\rangle$ as $L_+ |\Delta, 0\rangle = \sqrt{2\Delta} |\Delta, 1\rangle$.

(**2**) Discrete series $\mathcal{R}_\Delta^-$ are also generated by the states $|\Delta, N\rangle$ and can be constructed as follows

$$\begin{aligned}
L_- |\Delta, N\rangle &= -\sqrt{(N+1)(N+2\Delta)} \, |\Delta, N+1\rangle \\
L_+ |\Delta, N\rangle &= -\sqrt{N(N+2\Delta-1)} \, |\Delta, N-1\rangle \\
L_0 |\Delta, N\rangle &= -(N+\Delta) \, |\Delta, N\rangle \\
\mathcal{C}_2 |\Delta, N\rangle &= \Delta(\Delta-1) \, |\Delta, N\rangle
\end{aligned} \tag{3.4}$$

from which we can compute:

(**i**) The norm $< \Delta, N|L_+L_-|\Delta, N >$ giving $(N+1)(N+2\Delta)$; and the homologue $< \Delta, N|L_-L_+|\Delta, N >$ given by $N(N+2\Delta-1)$.

(**ii**) Conversely to the $\mathcal{R}_\Delta^+$ representation, $\mathcal{R}_\Delta^-$ is bounded from above with the constraint $L_+|\Delta, N\rangle = 0$ requiring $N(N+2\Delta-1) = 0$.

Analogously, if we impose $N = 0$, the highest weight state $|\Delta, 0\rangle$ annihilated by $L_+$ satisfies the relations

$$
\begin{aligned}
L_+|\Delta, 0\rangle &= 0 \\
L_0|\Delta, 0\rangle &= -\Delta|\Delta, 0\rangle \\
\mathcal{C}_2|\Delta, 0\rangle &= \Delta(\Delta-1)|\Delta, 0\rangle
\end{aligned}
\tag{3.5}
$$

with $L_-$ action given by $L_-|\Delta, 0\rangle = -\sqrt{2\Delta}|\Delta, -1\rangle$.

Notice that the two discrete representations $\mathcal{R}_\Delta^+$ and $\mathcal{R}_\Delta^-$ are isomorphic; the isomorphism $\iota : \mathcal{R}_\Delta^+ \to \mathcal{R}_\Delta^-$ is given by the 1:1 correspondence $\iota(L_n) = -L_{-n}$ as manifestly exhibited by the relations (3.2) and (3.4).

## 3.2 Higher spin AdS$_3$ gravity

Focussing on HS- BTZ black holes with rank 2 symmetries of eq(2.1), the gauge theory is described by the 3D HS gravity action $\mathcal{S}_0^{\text{GRAV}}$ given in terms of two copies of Chern-Simons (CS) fields $A$ and $\tilde{A}$ as follows [19, 20]

$$
\mathcal{S}_0^{\text{GRAV}} = \frac{\text{k}}{4\pi}\int tr(AdA + \frac{2}{3}A^3) - \frac{\tilde{\text{k}}}{4\pi}\int tr(\tilde{A}d\tilde{A} + \frac{2}{3}\tilde{A}^3)
\tag{3.6}
$$

with CS level $\tilde{\text{k}} = \text{k}$. This positive integer number is related to the AdS$_3$ radius and the 3D Newton coupling constant like $\text{k} = l_{\text{AdS}_3}/(4G_N)$. Being a discrete relation, this quantity can be imagined as a quantization relation of the 3D Newton constant expressed like $G_N^{[\text{k}]} = l_{\text{AdS}_3}/(4\text{k})$, showing in turns that $G_N^{[1]} = l_{\text{AdS}_3}/4$.

The conversion to the metric formulation is quite straightforward and mainly based on expressing both the dreibein $E_\mu$ and the spin connection $\Omega_\mu$ in terms of the two CS gauge potentials $A_\mu$ and $\tilde{A}_\mu$ as follows

$$
\begin{aligned}
G_{\mu\nu} &= \tfrac{1}{2}Tr(E_\mu E_\nu) \\
\Phi_{\mu_1\dots\mu_s} &= Tr\left(E_{(\mu_1}\dots E_{\mu_s)}\right) \\
E_\mu &= A_\mu - \tilde{A}_\mu \\
\Omega_\mu &= A_\mu + \tilde{A}_\mu
\end{aligned}
\tag{3.7}
$$

Notice also that here the 1-form gauge connections $A$ and $\tilde{A}$ as well as the $E$ and $\Omega$ are non abelian 3D fields; they are valued in the Lie algebra of the gauge symmetry $\mathcal{G}_{hs} \times \tilde{\mathcal{G}}_{hs}$ and satisfy the Grumiller-Riegler (GR) boundary conditions [32] for a more general set-up. The field equations of motion of (3.6) are given by $\mathcal{F}_{\mu\nu} = 0$ and $\tilde{\mathcal{F}}_{\mu\nu} = 0$ where the $\mathcal{F}_{\mu\nu}$ and $\tilde{\mathcal{F}}_{\mu\nu}$ are the gauge fields strengths reading as $\partial_\mu A_\nu - \partial_\nu A_\mu + [A_\mu, A_\nu]$ and $\partial_{[\mu} \tilde{A}_{\nu]} + [\tilde{A}_\mu, \tilde{A}_\nu]$.

Because of the vanishing value of the gauge field strengths, gauge invariants similar to the 4D electric field $\mathbf{E}_{u(1)}$ of (3.6) and the associated electric charge $q = \int_{\mathbb{S}^2} \mathbf{E}_{u(1)}.d\boldsymbol{\sigma}$ are unavailable in the higher spin AdS$_3$ gravity. Instead, there are alternative gauge invariants given by (**i**) the Wilson loops

$$\mathcal{W}_{\mathcal{R}}\left[\gamma\right] = Tr_{\mathcal{R}}\left[P \exp\left(\int_\gamma \boldsymbol{A}\right) P \exp\left(\int_\gamma \tilde{\boldsymbol{A}}\right)\right] \tag{3.8}$$

with $\mathcal{R}$ being a representation of the gauge symmetry, $\gamma$ a loop in AdS$_3$, $\boldsymbol{A}$ as well as $\tilde{\boldsymbol{A}}$ some gauge connections expanding as $A_\mu dx^\mu$ and $\tilde{A}_\mu dx^\mu$. And (**ii**) topological defects given by line operators constructed as [33, 34]

$$\mathcal{W}_{\mathcal{R}}\left[y_i, y_f\right] = \langle U_i | Tr_{\mathcal{R}}\left[P \exp\left(\int_{\Upsilon_{if}} \boldsymbol{A}\right) P \exp\left(\int_{\Upsilon_{if}} \tilde{\boldsymbol{A}}\right)\right] |U_f\rangle \tag{3.9}$$

where $(y_i, y_f)$ are the end points of the curve $\Upsilon_{if}$ parameterised by $y$ and where $U(y)$ is a probe field on $\Upsilon_{if}$ with boundary condition $U(y_i) = U(y_f) = I_{id}$. As illustrations, we give the expansion of the potential $A_\mu$ for the SL(3,$\mathbb{R}$) and G$_2$ models in the higher spin basis. For SL(3,$\mathbb{R}$), we have the following splitting

$$\begin{aligned}
\text{SL(3,}\mathbb{R}\text{)} \quad : \quad A_\mu &= \sum_{m_1=-1}^{1} \mathcal{A}_\mu^{m_1} W_{m_1}^{(1)} + \sum_{m_2=-2}^{2} \mathcal{W}_\mu^{m_2} W_{m_2}^{(2)} \\
&:= \sum_{m=-1}^{1} \mathcal{A}_\mu^m L_m + \sum_{n=-2}^{2} \mathcal{W}_\mu^n W_n
\end{aligned} \tag{3.10}$$

with the commutation relations [36]

$$\begin{aligned}
[L_i, L_j] &= (j-i) L_{i+j} \\
[L_i, W_m] &= (m-2i) W_{i+m} \\
[W_n, W_m] &= \tfrac{1}{3}(n-m)(2m^2 + 2n^2 - mn - 8) L_{n+m}
\end{aligned} \tag{3.11}$$

Similarly for G$_2$, we can write

$$\begin{aligned}
\text{G}_2 \quad : \quad A_\mu &= \sum_{m_1=-1}^{1} \mathcal{A}_\mu^{m_1} W_{m_1}^{(1)} + \sum_{m_5=-5}^{5} \mathcal{W}_\mu^{m_5} W_{m_5}^{(5)} \\
&:= \sum_{m=-1}^{1} \mathcal{A}_\mu^m L_m + \sum_{n=-5}^{5} \mathcal{W}_\mu^n W_n
\end{aligned} \tag{3.12}$$

with

$$
\begin{aligned}
[L_i, L_j] &= (j - i) L_{i+j} \\
[L_i, W_m] &= (m - 5i) W_{i+m} \\
[W_n, W_m] &= f^{(1,5)}_{m,n|2} L_{n+m}
\end{aligned}
\tag{3.13}
$$

where $f^{(1,5)}_{m,n|2}$ are constant structures obtained by solving the Jacobi identities. Notice that in the HS- basis, the 8 generators of SL(3,$\mathbb{R}$) are split into two blocks 3+5 given by: (**i**) the three $W^{(1)}_{m_1}$ with label $m_1 = 0, \pm 1$; they are just the usual generators $L_m$ of the principal SL(2,$\mathbb{R}$). (**ii**) The five $W^{(2)}_{m_2} \equiv W_n$ with index $m_2 = n = 0, \pm 1, \pm 2$; they generate the coset space SL(3,$\mathbb{R}$)/SL(2,$\mathbb{R}$). Quite similar relations can be written for the 14 generators of $G_2$ that split as $3 + 11$.

As far as these types of HS- expansions are concerned, notice the following features depicted for the case of SL(3,$\mathbb{R}$) model: (**i**) The commutation relations of sl(3,$\mathbb{R}$) in the HS basis can be presented in a condensed form as follows

$$
\left[ W^{(j)}_{m_j}, W^{(k)}_{n_k} \right] = \sum_{r_1=-1}^{1} f^{(j,k)}_{n_k,m_j|1} \delta^{r_1}_{m_j+n_k} W^{(1)}_{r_1} + \sum_{r_2=-2}^{2} f^{(j,k)}_{n_k,m_j|2} \delta^{r_2}_{m_j+n_k} W^{(2)}_{r_2}
\tag{3.14}
$$

with the constant structures $f^{(j,k)}_{m_j,n_k|s}$ given by

$$
\begin{aligned}
f^{(1,1)}_{n_1,m_1|1} &= m_1 - n_1 \\
f^{(1,2)}_{n_2,m_1|2} &= m_1 - 2n_2 \\
f^{(2,2)}_{n_2,m_2|2} &= \tfrac{1}{3} (n_2 - m_2)(2m_2^2 + 2n_2^2 - m_2 n_2 - 8)
\end{aligned}
\tag{3.15}
$$

In general, we can express these commutations in a shorter form like

$$
\left[ W^{(\tau)}_{m_\tau}, W^{(\sigma)}_{n_\sigma} \right] = \sum_{\upsilon} \sum_{r_\upsilon} f^{(\tau,\sigma)}_{n_\sigma,m_\tau|\upsilon} \delta^{r_\upsilon}_{m_\tau+n_\sigma} W^{(\upsilon)}_{r_\upsilon}
\tag{3.16}
$$

(**ii**) Higher spin theories are characterised by the spins- s of the principal SL(2,$\mathbb{R}$) within SL(3,$\mathbb{R}$); it is defined by the usual commutation relations (2.4) where we have set $L_m = W^{(1)}_{m_1}$. As such, it is interesting to use the formal decomposition

$$
SL(3, \mathbb{R}) = SL(2, \mathbb{R}) \ltimes \frac{SL(3, \mathbb{R})}{SL(2, \mathbb{R})}
\tag{3.17}
$$

to split the gauge potentials $A_\mu$ and $\tilde{A}_\mu$ as follows

$$
\begin{aligned}
A_\mu &= A^{sl_2}_\mu + A^{sl_{3/2}}_\mu \\
\tilde{A}_\mu &= \tilde{A}^{sl_2}_\mu + \tilde{A}^{sl_{3/2}}_\mu
\end{aligned}
\tag{3.18}
$$

# 4  Derivation of the HS Swampland conjecture

In this section, we target the derivation of the HS Swampland conjecture in AdS$_3$ (2.12) which can be articulated as in the following statement:

**Higher spin Swampland conjecture in AdS$_3$**

*A higher spin BTZ black hole solution of 3D topologically massive gravity with a negative cosmological constant $\Lambda < 0$ should be able to discharge by emitting super-extremal higher spin particles with mass* $M_{hs}$ *and charge* $Q_{hs}$ *such that*

$$M_{hs} \leq \alpha_3 Q_{hs} g_{hs} M_{Pl} \tag{4.1}$$

*where* $g_{hs}$ *is the higher spin gauge coupling and* $\alpha_3$ *is some constant that we set as* $\alpha_3 = \sqrt{2}$.

The constraint (4.1) bears a mighty resemblance to the inequality (2.16) regulating the decay of charged 4D black holes,

$$
\begin{array}{ccc}
\text{3D HS-BTZ} & \leftrightarrow & \text{4D charged BH} \\
M_{hs} \;\; \leq \;\; \sqrt{2} Q_{hs} g_{hs} M_{Pl} & \leftrightarrow & \boldsymbol{m} \;\; \leq \;\; \sqrt{2} q g_{U(1)} M_{Pl}
\end{array}
\tag{4.2}
$$

but instead of the abelian U(1) parameters, we must determine the higher spin $M_{hs}$, $Q_{hs}$ and $g_{hs}$ quantities for the $\mathcal{G}_{hs}$ symmetry. For this purpose, we first promote the 3D gravity theory described by the field action $\mathcal{S}_0^{GRAV}$ to a higher spin topologically massive AdS$_3$ gravity [6, 8] in order for our, as of yet, massless higher spin states to acquire mass. This is accomplished by adding to the gauge field action (3.6) a gravitational Chern-Simons term $\mathcal{S}_1^{GRAV}$ given by [6]-[9]

$$\mathcal{S}_1^{GRAV} = \frac{M_{Pl}}{2\mu} \int_{\mathcal{M}_{3D}} Tr\left(\Gamma d\Gamma + \frac{2}{3}\Gamma^3\right) \tag{4.3}$$

where $\Gamma$ is the Christoffel symbol and where $\mu$ is a massive parameter. The mass $M_{hs}$ of the higher spin super extremal states is therefore a function of the parameter $\mu$ and the conformal spin s such that $M_{hs} = M(s, \mu)$. Following the conjecture of [4, 37, 38], $M_{hs}$ can be formulated as

$$M_{hs}^2 = \frac{1 + \mu l_{AdS_3}}{l_{AdS_3}^2}(s-1)\left[(s-1)\mu l_{AdS_3} + (s+1)\right] \tag{4.4}$$

This is a remarkable relation that can be put into a covariant form using observables of the principal SL(2,$\mathbb{R}$) symmetry of the higher spin theory. In fact, by putting $M_{AdS_3} = 1/l_{AdS_3}$ into the above $M_{hs}^2$ relation and after rearranging the terms, we end up with the distinguishable expression

$$M_{hs}^2 = (M_{AdS_3} + \mu)^2 s(s-1) + (M_{AdS_3}^2 - \mu^2)(s-1) \tag{4.5}$$

more thoroughly investigated below. The relationship between this conjectured mass formula and those $M^2_{\Delta,N_+}$ and $M^2_{\Delta,N_-}$ given by (2.11), associated with the two unitary representations $\mathcal{R}^+_\Delta$ and $\mathcal{R}^-_\Delta$, will be commented in subsection 4.2. Before that, let us see how the swampland constraint relation can be derived from (4.5).

## 4.1 From eq(4.5) towards eq(4.1)

Using the relation $s = 1 + j$, linking the values of the conformal spins- s of the higher spin AdS$_3$ gravity to the isospin j representation weights of SL(2,$\mathbb{R}$), the above conjectured mass relation $M^2_{hs}$ becomes

$$M^2_{hs} = (M_{AdS_3} + \mu)^2 j (j + 1) + (M^2_{AdS_3} - \mu^2) j$$
$$= \left(\frac{1 + \mu l_{AdS_3}}{l_{AdS_3}}\right)^2 j (j + 1) + \left(\frac{1 - \mu^2 l^2_{AdS_3}}{l^2_{AdS_3}}\right) j \qquad (4.6)$$

exhibiting two well known quantum numbers of SL(2,$\mathbb{R}$) representations namely the weight $j$ and the Casimir $j (j + 1)$. Moreover, since $j \geq 1$ due to the condition $s \geq 2$, we have the property $j (j + 1) > j$ implying that the dominant term in the $M^2_{hs}$ formula is given by the block term $(M_{AdS_3} + \mu)^2 j (j + 1)$. Furthermore, we can note two additional valuable features:

(i) In the region of the parameter space of the higher spin theory where $M^2_{AdS_3} - \mu^2$ is negative definite ( i.e: $1 - \mu^2 l^2_{AdS_3} < 0$); we have

$$\mu^2 > M^2_{AdS_3} \qquad \Leftrightarrow \qquad \mu^2 > \frac{1}{l^2_{AdS_3}} \qquad (4.7)$$

and then the mass formula (4.6) induces the following inequality

$$M^2_{hs} < (M_{AdS_3} + \mu)^2 j (j + 1) \qquad \Leftrightarrow \qquad M^2_{hs} < \left(\frac{1 + \mu l_{AdS_3}}{l_{AdS_3}}\right)^2 j (j + 1) \qquad (4.8)$$

which corresponds precisely to (4.1); thus offering a natural candidate for the swampland conjecture regarding HS topological AdS$_3$ massive gravity.

(ii) For the critical value $\mu^2 = \mu^2_c = M^2_{AdS_3}$, the block term $(M^2_{AdS_3} - \mu^2_c) j$ in (4.6) vanishes; and the mass formula $M^2_{hs}$ in (4.6) is equal to $(M^2_{hs})_c = 4M^2_{AdS_3} j (j + 1)$. So, using $\mu^2 \simeq M^2_{AdS_3} + \delta\mu^2$ with positive $\delta\mu^2$, eq(4.6) becomes

$$M^2_{hs} = 4M^2_{AdS_3} j (j + 1) - (\delta\mu^2) j \qquad (4.9)$$

thus leading to the inequality

$$M^2_{hs} \leq 4M^2_{AdS_3} j (j + 1) \qquad \Leftrightarrow \qquad M^2_{hs} \leq \frac{4}{l^2_{AdS_3}} j (j + 1) \qquad (4.10)$$

In comparison with (4.1) stipulating $M_{hs}^2 \leq 2Q_{hs}^2 g_{hs}^2 M_{Pl}^2$, one can deduce the expressions of both the charge $Q_{hs}$ and the coupling constant $g_{hs}$; they are given by

$$Q_{hs}^2 = j\,(j+1) = s\,(s-1) \tag{4.11}$$

$$g_{hs}^2 = \frac{2M_{AdS_3}^2}{M_{Pl}^2} = \frac{2}{M_{Pl}^2 l_{AdS_3}^2} \tag{4.12}$$

with higher spin $s = 1 + j$. The expression of the coupling constant can be presented otherwise by using the CS level relation $k = l_{AdS_3}/(4G_N)$, which gives

$$g_{hs}^2 = \frac{1}{8k^2 G_N^2 M_{Pl}^2} \tag{4.13}$$

where the dependence on the Chern-Simons coupling k, the Newton constant $G_N$ as well as Planck mass $M_{Pl}$ is exhibited. As these constant are interconnected, we can further unclutter the expression by using the relation $M_{Pl}G_N = 1/(8\pi)$ to showcase that $g_{hs}$ is merely the inverse of the Chern-Simons k:

$$g_{hs}^2 = \left(8\pi^2\right)/k^2 \tag{4.14}$$

## 4.2 Refining eqs(4.11-4.12) and the super-extremal tower

The emergence of quantum numbers of the SL(2,$\mathbb{R}$) representations in the conjectured mass formula (4.6) makes one ponder about other hidden facets of $M_{hs}^2$. Below, we give two interesting features allowing to refine the eqs(4.11-4.12):

The first feature concerns the algebraic interpretation of the expression $M_{hs}^2$ (4.6); in fact, $M_{hs}^2$ can be perceived as the eigenvalue of a mass operator $\hat{M}^2$ acting on the quantum particle states $|\Delta, N\rangle$ emitted by the HS-BTZ black hole as follows

$$\hat{M}^2 |\Delta, N\rangle = M_{hs}^2 |\Delta, N\rangle \tag{4.15}$$

with positive inetegers $\Delta$ and N. Acting on these quantum states $|\Delta, N\rangle$ by the mass operator

$$\hat{M}^2 = (M_{AdS_3} + \mu)^2\, \mathcal{C}_2 + \left(M_{AdS_3}^2 - \mu^2\right) L_0 \tag{4.16}$$

we get its eigenvalues in terms of the quantum numbers $\Delta$ and $N$; they read as follows

$$M_{\Delta,N}^2 = (M_{AdS_3} + \mu)^2\, \Delta\,(\Delta - 1) + \left(M_{AdS_3}^2 - \mu^2\right)(\Delta + N) \tag{4.17}$$

Because $N \in \mathbb{N}$, the quantum states $|\Delta, N\rangle$ define an infinite tower of states candidates for the emitted particles of the HS-BTZ black hole. In this regard, recall that in AdS$_3$ one must upgrade the mild WGC to stronger forms like the lattice WGC of [17]. The

refined HSC is given by the tower WGC [39] occupied by super extremal higher spin states (4.17) fulfilling the mass to charge constraint (4.1).

The second feature regards the HS Swampland conjecture (4.2) namely $\text{M}_{\text{hs}} \leq \sqrt{2}\text{Q}_{\text{hs}}g_{\text{hs}}M_{\text{Pl}}$. This inequality puts a constraint on the appropriate unitary representation of $SL_2$ where the tower of super extremal particle states $|\Delta, N\rangle$ emitted by the HS-BTZ black hole resides. Because $M_{\text{AdS}_3}^2 - \mu^2$ has an indefinite sign, we can distinguish three types of mass operators according to the value of $\mu^2$ compared to $M_{\text{AdS}_3}^2$. We have

$$
\begin{array}{rcccl}
(a) & : & \mu^2 & > & M_{\text{AdS}_3}^2 \\
(b) & : & \mu^2 & = & M_{\text{AdS}_3}^2 \\
(c) & : & \mu^2 & < & M_{\text{AdS}_3}^2
\end{array}
\tag{4.18}
$$

these three phases are very common in the study of TMG theories [6, 7, 40, 41]. In fact by considering the additional gravitational CS term (4.3), the massive HS gravity theory develops a diffeomorphism anomaly given by the difference between the right $c_+$ and the left $c_-$ central charges

$$
c_\pm = \frac{3l_{\text{AdS}_3}}{G_N} \left( 1 \pm \frac{1}{\mu l_{\text{AdS}_3}} \right)
\tag{4.19}
$$

leading to

$$
\frac{1}{\mu} = \frac{G_N}{6} (c_+ - c_-)
\tag{4.20}
$$

The value of $\mu$ is therefore a measure of the violation of parity in TMG. Additionally, one must note that the central charges are positive definite when $\frac{1}{\mu l_{\text{AdS}_3}} \leq 1$. As for the critical value $\mu l_{\text{AdS}_3} = 1$, it implies the vanishing of the central charges $c_-$ and the resulting TMG theory was shown to be dual to a logarithmic CFT [37].

For all three phases (4.18), the mass operator takes the following forms

$$
\begin{array}{rcccl}
(a) & : & \hat{\text{M}}_-^2 & = & (M_{\text{AdS}_3} + \mu)^2 \, \mathcal{C}_2 - \left| M_{\text{AdS}_3}^2 - \mu^2 \right| L_0 \\
(b) & : & \hat{\text{M}}_0^2 & = & 4M_{\text{AdS}_3}^2 \mathcal{C}_2 \\
(c) & : & \hat{\text{M}}_+^2 & = & (M_{\text{AdS}_3} + \mu)^2 \, \mathcal{C}_2 + \left| M_{\text{AdS}_3}^2 - \mu^2 \right| L_0
\end{array}
\tag{4.21}
$$

Acting by these operators on the particle states $|\Delta, N\rangle$, we obtain the eigenvalues

$$
\begin{array}{rcl}
(\text{M}_-^2)_{\Delta,N} & = & (M_{\text{AdS}_3} + \mu)^2 \, \Delta (\Delta - 1) - \left| M_{\text{AdS}_3}^2 - \mu^2 \right| (\Delta + N) \\
(\text{M}_0^2)_{\Delta,N} & = & 4M_{\text{AdS}_3}^2 \Delta (\Delta - 1) \\
(\text{M}_+^2)_{\Delta,N} & = & (M_{\text{AdS}_3} + \mu)^2 \, \Delta (\Delta - 1) + \left| M_{\text{AdS}_3}^2 - \mu^2 \right| (\Delta + N)
\end{array}
\tag{4.22}
$$

which for $\Delta > 1$, they obey the inequalities

$$
\begin{array}{rcl}
(\text{M}_-^2)_{\Delta,N} & < & (M_{\text{AdS}_3} + \mu)^2 \, \Delta (\Delta - 1) \\
(\text{M}_0^2)_{\Delta,N} & = & 4M_{\text{AdS}_3}^2 \Delta (\Delta - 1) \\
(\text{M}_+^2)_{\Delta,N} & > & (M_{\text{AdS}_3} + \mu)^2 \, \Delta (\Delta - 1)
\end{array}
\tag{4.23}
$$

showing that $(\text{M}_0^2)_{\Delta,N}$ is a critical mass. This feature allows to think about the $(\text{M}_-^2)_{\Delta,N}$ inequality as follows

$$(\text{M}_-^2)_{\Delta,N} \leq 4M_{\text{AdS}_3}^2 \Delta(\Delta - 1) \qquad \Leftrightarrow \qquad \text{M}_{\text{hs}} \leq \sqrt{2}\text{Q}_{\text{hs}}g_{\text{hs}}M_{\text{Pl}} \qquad (4.24)$$

from which we deduce the HS charge $\text{Q}_{\text{hs}}$ and the HS coupling constant $g_{\text{hs}}$ supported by the representation theory,

$$\text{Q}_{\text{hs}} = \sqrt{\Delta(\Delta - 1)} \qquad \Leftrightarrow \qquad g_{\text{hs}} = \sqrt{2}\frac{M_{\text{AdS}_3}}{M_{\text{Pl}}} \qquad (4.25)$$

Notice finally that expressing eqs(4.22) as

$$(\text{M}_\pm^2)_{\Delta,N} = (M_{\text{AdS}_3} + \mu)^2 \Delta(\Delta - 1) \pm \left|M_{\text{AdS}_3}^2 - \mu^2\right|(\Delta + N) \qquad (4.26)$$

we see that these masses $(\text{M}_\pm^2)_{\Delta,N}$ are intimately related to the unitary $\text{SL}_2$ representations $\mathcal{R}_\Delta^\pm$. The tower of states fulfilling HS Swampland conjecture (4.24) is then given by the quantum states of $\mathcal{R}_\Delta^-$.

## 5 Piecing HSC in the WGC framework

The weak gravity conjecture is one of the seminal ideas in the swampland program, and may very well be the most properly argued swampland criteria. It has been studied in numerous settings with various parametrisations and configurations, giving many formulations that differ both in their assumptions as well as in their regime of applicability, for an extensive review refer to [42]. Pertaining to our concern, we will briefly look over some of its statements and implications for AdS theories.

### 5.1 WGC in AdS background

A prerequisite of any potential WGC formulation in a curved $\text{AdS}_d$ space is the possibility to recover the usual bound of the flat space once the curvature $l_{AdS_\text{d}} \to \infty$ [42]. Unfortunately, a general AdS formulation of the WGC is still a pending issue. However, there are many proposals like the one in [43]:

$$\frac{\delta^2}{l_{AdS_\text{d}}^2} \leq \frac{d-2}{d-3}\frac{e^2q^2}{G_N^2} \qquad (5.1)$$

where $\delta$ is the conformal scaling dimension related to the mass $\mathbf{m}$ via

$$\delta = \frac{d-1}{2} + \sqrt{\frac{(d-1)^2}{4} + l_{AdS_\text{d}}^2\mathbf{m}^2} \qquad (5.2)$$

In addition to the bound (5.1) having the d=3 singularity, it is not satisfied for all CFTs and it is unclear why this particular condition is most likely to hold universally [42]. Another Anti de Sitter WGC reformulation is given by the charge convexity conjecture [44], it imposes bounds in terms of binding energy using the lowest dimension operator of the associated CFT. Although the convex charge constraint is believed to be more general than the WGC, we disregard it as it differs from the usual statements motivated by black holes decay or long range forces.

To overcome the triviality of the constraint (5.1) for d=3, there is an alternative method that exploits tools of the $AdS_3/CFT_2$ correspondence. In [17] and more generally in [18], the weak gravity conjecture was indeed derived using a conformal approach by demanding the partition function of the boundary $CFT_2$ to be modular invariant. In a disjointed setting [18], where the gravitational and gauge sectors are distinct by considering 3D gravity in addition to a U(1) gauge field, it is possible to establish a constraint on the conformal dimension of the lightest charged state as follows [18],

$$\delta - \delta_{\text{VAC}} \simeq \frac{c}{6} + \frac{3}{2\pi} + \mathcal{O}\left(\frac{1}{c}\right) \tag{5.3}$$

This bound is not optimal, and can be enhanced via additional symmetries. In fact, for 2D supersymmetric CFT with $\mathcal{N} = (1,1)$ supercharges, the constraint (5.3) on the conformal weight improves to $\delta \simeq 1 + O(1/c)$.

However, the constraint (5.3) isn't suitable for HS-TMG as it doesn't consider charged higher spin fields and only concerns U(1) charges.

| Formulations | AdS background | D=3 | BH solution | Massive HS fields | HS charge |
|---|---|---|---|---|---|
| WGC in AdS [43] | x | - | x | - | - |
| Convex Charge Constraint [44] | x | x | - | - | - |
| WGC in $AdS_3$ [17] | x | x | x | - | - |

(5.4)

## 5.2 Beyond electric U(1) charges

There are other formulations of the WGC that experimented with parameters beyond the typical electric U(1) charges. For instance, the so called spinning weak gravity conjecture [45] where quantum or higher derivative corrections lead to perturbed (BTZ) black holes

obeying a rotating version of the WGC that follows from the holographic c-theorem. Another interesting case is the causality bounds on higher spin particles coupled to stringy gravity in 4D [46]. In fact, in order for a 4D gravitational theory coupled to a tower of higher spin states to be causal, a WGC-like constraint must be imposed on the lightest HS particle. The 4D causality bound is reminiscent of the spin-2 conjecture requiring a cutoff on gravitational theories with massive higher spin fields [47].

| Formulations | AdS background | D=3 | BH solution | Massive HS fields | HS charge |
|---|---|---|---|---|---|
| WGC in AdS [43] | x | - | x | - | - |
| Convex Charge Constraint [44] | x | x | - | - | - |
| WGC in AdS$_3$ [17] | x | x | x | - | - |
| A spinning WGC [45] | x | x | x | - | - |
| HS causality [46] | - | - | x | x | - |
| Our HSC proposal | x | x | x | x | x |

$$(5.5)$$

As evidenced, the HSC addresses a setting with a particular configuration to investigate the WGC. We derive a WGC-like constraint for black hole solutions of higher spin topological massive gravity carrying higher spin charges. The HSC stems from the core SL(2) algebraic representations and provides a constraint on the HS fields masses and charges to regulate the discharge of the HS BTZ solutions.

Exploring swampland conjectures from the lens of holographic theories has been of great interest recently. While we mainly focused on the WGC, there is a substantial body of work relating the swampland distance conjecture to higher spin theories as in [48] and the ensuing [49, 50]. For instance in [48], it has been proposed that at infinite distances all theories possess an emergent HS symmetry in such a manner that certain proprieties of the conformal manifolds can be written as a function of the HS spectrum.

# 6 Conclusion

In this paper, we investigated a well motivated inquiry regarding the discharge of higher spin BTZ black holes in a higher spin topological massive gravity setting with Chern-Simons formulation based on rank-2 higher spin gauge symmetries. We proposed a

higher spin Swampland conjecture to regulate the emission of super-extremal higher spin particles given by an upper bound on their mass to charge ratio.

En route to derive the higher spin swampland conjecture, we first established a correspondence between the massive higher spin AdS$_3$ models and effective gauge theories coupled to D-gravity (EFF$_D$) to hypothesize a formulation of the swampland constraint for higher spin BTZ black holes. Exploiting the principal SL(2,$\mathbb{R}$) of the higher spin gauge symmetry, we constructed the charge (3.1) and the mass (4.16) operators as well as their eigenvalues (4.22, 4.11). We also computed the higher spin gauge coupling constant (4.12) and showcased its relation to the inverse of the Chern-Simons level k (4.14).

Furthermore by using the infinite dimensional unitary representations, particularly the discrete series $\mathcal{R}_\Delta^-$, we built a tower of higher spin states (4.26) occupied by the emitted higher spin particles in accordance with the lattice refinement required for the AdS$_3$ space. We must note that the mass operator leading to the tower of higher spin states ensues from the phase $\mu^2 > M_{\text{AdS}_3}^2$ assuring the positivity of the central charges (4.19) as well as the unitarity of the CFT.

On a final note, we discussed the various WGC formulations especially for AdS backgrounds in different settings to place the higher spin swampland conjecture within the WGC framework as a way to emphasize the pertinence of our work regarding recent advancements in the swampland program. Overall, the antecedent results may imply several interpretations:

(**i**) The inclusivity of topological massive gravity within the general Landscape of consistent quantum gravitational theories.

(**ii**) Particularly, the established link between the higher spin conjecture and the WGC constraint conveys the validity of the later for topological massive higher spin gravitational models.

(**iii**) The existence of the tower of higher spin states is strongly supported by algebraic properties of the core SL(2,$\mathbb{R}$) of the HS gravity namely the discrete infinite unitary representation $\mathcal{R}_\Delta^-$.

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
