# Peer review of "Higher spin swampland conjecture for massive AdS$_{3}$ gravity"

_SciPost Physics_

## Round 1 · Referee Report · Anonymous (Referee 1) · 2024-10-27

Report

The authors conjecture a possible extension of the Weak Gravity Conjecture (WGC) to higher spin massive topological gravity on AdS3 backgrounds by considering the decay of the higher spin generalization of BTZ black holes in these theories. To develop their conjecture, they need to identify the appropriate states that conjecturally mediate the discharge of these black holes, and discuss their mass, charge, and coupling. They argue for this identification by using a Chern-Simons formulation of higher-spin AdS3 gravity, supplemented by a mass term.

The questions studied in this paper are relevant for extending the applicability of one of the most consequential and well-studied conjectures in the swampland program, and are thus a valuable addition to the swampland literature.

Before recommending the paper for publication, I would like to ask a few clarification questions to the authors:

  1. Could the authors provide a short self-contained review of the massive formulation the higher-spin theory? Specifically, it would be beneficial for the reader if the authors could review the properties of the action (4.3) and, in particular, how formula (4.4) is obtained. This is particularly relevant in light of the fact that (4.4) is the crucial starting point for their derivation of the inequalities in section 4.1.

  2. It would also beneficial if the authors could expand their discussion in section 5 by commenting on the differences between the topological nature of the theory they are considering, with local degrees of freedom added by the mass deformation, with the more standard setup of the WGC with local U(1) degrees of freedom.

  3. The authors mention in the introduction that for AdS3 weak forms of the WGC are not enough to ensure black hole decay, thus requiring a lattice version of the WGC. Is it obvious that this same argument directly applies to the higher-spin case? Could it be that the self-interacting particle condensates might not form in the class of theories the authors are considering? It can be helpful for the readers if the authors could expand on this point, discussing possible difference that might arise in higher-spin theories.

Recommendation

Ask for minor revision

  • validity: -
  • significance: -
  • originality: -
  • clarity: -
  • formatting: -
  • grammar: -

Author:  Rajae Sammani  on 2024-11-01  [id 4925]

(in reply to Report 1 on 2024-10-27)
Category:
answer to question

Please see the attached file for clarifications on the points raised in the report.

Attachment:

1Nov-To-Referee-SciPost.pdf

---

## Round 1 · Referee Report · Anonymous (Referee 2) · 2024-11-19

Strengths

1- Interesting idea on a topic of current interest

Weaknesses

1- A conjecture that depends on a previous conjecture 2- Unclear argument for identification of charge

Report

This paper considers the possibility that the weak gravity conjecture might apply to higher spin theories as well. This is an interesting idea worth considering.

However, I am not quite sure whether the authors provided convincing evidence. Formula (4.4) is a conjecture presented in (16) of Ref.[4], based on the spin-two results in [37,38] and on some other papers (1106.5141, 1107.0915, 1107.2063). As the other referee is pointing out, it would be beneficial to review critically the evidence behind this conjecture, before formulating a further conjecture that depends upon it.

Moreover, after massaging this into (4.10), the authors identify that the two factors are the coupling constant and the charge. What is the evidence for this identification? If we are just being guided by what we would like to be true, namely (4.1), then it seems to me that we are not really providing an argument.

In my opinion, it would be important to provide some independent argument (not based on what we would like to prove) that $j(j+1)=s(s-1)$ can indeed be considered a charge. Such evidence might be provided for example by the behavior of higher spin black holes.

Requested changes

1- Review of why (4.4) might be true 2- Independent argument to justify why $s(s-1)$ can be thought of as charge

Recommendation

Ask for major revision

---

## Round 2 · Referee Report · Anonymous (Referee 2) · 2025-1-1

Strengths
1- addressed almost all points raised by referees
Weaknesses
1- one crucial point remains a little weak
Report
Below (4.23), the authors also tried addressing my other question, about why $s(s-1)$ can be interpreted as a charge. I thank them for their effort. But I have to confess that I am not sure I follow the logic here: in the last line of p. 16, "an extremal higher spin BTZ black hole is then a black hole with mass equal to $M_{\rm hs}^2 = \left(\frac1{l_{\rm AdS_3}}+\mu \right)^2 j(j+1)$", where does this formula come from? Is this a circular argument?
I think the idea is roughly speaking to compare the extremality bound for Kerr–Newman to the expected WGC bound; for them to coincide, the $a^2$ term has somehow to coincide with $Q^2$.
My response to this is that $a$ here is the _angular momentum_ of the black hole, not quite the spin of the particles in the theory. I can see that probably the extremal Kerr–Newman in the higher spin theory might be similar to this one. But I think it might be possible to make this a lot sharper by comparing with existing literature. There is for example 1404.3305 (see for example the comment below (3.40) there), but a lot more has probably been done since then.
So as a final request I would urge the authors to rewrite the part below (4.23) a little more clearly, and to try to look whether this feature persists in actual higher spin theories. If this is hard, I would ask them to justify why.
Recommendation
Ask for minor revision

---

## Round 3 · Referee Report · Anonymous (Referee 1) · 2025-4-29

Report

I thank the authors for addressing the remaining point raised by both referees.

In the revised version, the authors have enhanced their discussion in section 4.1 by providing a better justification for their identification of $j(j+1)$ with $s(s-1)$. They accomplish this through a comparison between the charge term in super-extremality conditions for particle states in D-dimensional effective theories coupled to gravity and the notion of higher spin charge relevant to their case. The authors now also elaborate on how, in the Kerr-Newman scenario, one can go from bounds on particle states to bounds on black holes, while explaining why they believe a similar derivation in their context is too challenging to include in the present work and needs to be deferred to future research.

I believe the authors have now more effectively addressed the remaining issue with the manuscript, highlighting the limitations of their current analysis. I would encourage them to emphasize these points in their conclusions as well, reiterating the need for future work to fully address this matter. With these considerations in mind, I am happy to recommend this work for publication.

Recommendation

Publish (meets expectations and criteria for this Journal)

---

## Round 3 · Referee Report · Anonymous (Referee 2) · 2025-5-10

Report

I thank the authors for their efforts in clarifying the relation between charge and spin. It is true that settling the issue discussed in my previous report would probably require more work in extremality bounds for black holes in 3D higher spin massive gravity.

Recommendation

Publish (meets expectations and criteria for this Journal)

---

## Editorial Decision

accepted_in_target_journal